# REVISITING DENOISING AUTO-ENCODERS

**Luis Gonzalo Sanchez Giraldo**[*]
Department of Computer Science
University of Miami
Coral Gables, FL 33124, USA
lgsanchez@cs.miami.edu

## ABSTRACT

Denoising auto-encoders (DAE)s were proposed as a simple yet powerful way to obtain representations in an unsupervised manner by learning a map that approximates the clean inputs from their corrupted versions. However, the original objective function proposed for DAEs does not guarantee that denoising happens only at the encoding stages. We argue that a better representation can be obtained if the encoder is forced to carry out most of the denoising effort. Here, we propose a simple modification to the DAE's objective function that accomplishes the above goal.

## 1 INTRODUCTION

Auto-encoders (AE)s are unsupervised learning algorithms that capture structure in data by finding an representation of the inputs (encoding) from which they can be recovered at least approximately. By learning a transformation $G$ (encoding) from the input $x \in \mathcal{X}$ to $z = G(x) \in \mathcal{Z}$, the auto-encoder tries to capture the structure of the input. To guarantee the transformation $G$ preserves the information about $x$, a decoder $\tilde{G}^{-1}$ (an approximate inverse map) is also learned such that a measure of fidelity $\mathrm{E}[D(X, \tilde{X})]$ between the input and its reconstruction is optimized.

Learning non-linear encoding and decoding mappings has proved to be a non-trivial problem. For example, there is no consensus on what the dimensionality of the encoding space should be. On one side of the spectrum, AE architectures such as bottle-neck networks, map input data $x \in \mathcal{X} \subseteq \mathbb{R}^d$ to a lower dimensional space $\mathcal{Z} \subseteq \mathcal{R}^p$, and then map it back to input space $\mathcal{X}$ through dimensionality expansion. The intuition behind bottleneck networks is that the lower dimensional space forces the encoder to capture meaningful relations between the variables in the input space. While this is clear when one is restricted to linear mappings, the problem becomes less well-understood if non-linear mappings are allowed. The idea of using a low dimensional latent space has been recently employed in the context of modern generative models such as variational auto-encoders (VAE)s Kingma & Welling (2013) and generative adversarial networks (GAN)s Goodfellow et al. (2014).

On the other hand, AE architectures can contain over-complete representations where inputs are mapped to a high dimensional encoding spaces; much higher than the dimensionality of the input space ($\dim(\mathcal{X}) < \dim(\mathcal{Z})$). Over-complete representations have proved very useful in supervised learning. However, effective learning of this high dimensional mappings relies on specialized mechanisms that attempt to avoid trivial solutions or on the large amount of constraint imposed by the supervised task. Approaches such as sparse encoding, which can be applied to over-complete scenarios, use the concept of effective dimensionality. A sparsity constraint induces an active set of variables with an expected $L^0$ norm smaller than the input space dimensionality. Many sparse coding procedures require solving an optimization problem at inference time since the encoding mapping is not explicit. However, it has been shown that efficient inference can be made possible by training a nonlinear feed-forward network to mimic the output of a sparse encoding algorithm Ranzato et al. (2006) and more recently with the proposal of techniques that impose very strong sparsity constraints such as winner take all auto-encoders Makhzani & Frey (2015).

An important set of techniques that do not necessarily fall into either extreme in terms of dimensionality are based on the concept of robustness of the representation. A representation is said to be

---

[*]

robust if is able to retain information about the input even if the input or the representation undergo perturbations. For example, contractive auto-encoders (CAE)s Rifai et al. (2011a;b) penalize the sensitivity of the encoding map $G(x)$ to perturbations of the input $x$ by minimizing the expected Frobenius norm of the Jacobian $\mathrm{E}[J_G(X)]$ of $G$ while maximizing fidelity of the reconstruction. Within this category, learning representations by local denoising, or better known as de-noising auto-encoders, provide a very general way to carry out unsupervised learning. De-noising auto-encoders Vincent et al. (2008) were conceived as a very intuitive way to capture robustness in the encoding-decoding mapping by simply minimizing the error between the output reconstruction of the auto-encoder when a corrupted version of the input is feed to the map and the uncorrupted input. However, as already pointed out in Rifai et al. (2011a), the DAE objective minimizes the reconstruction error, therefore DAEs lack of explicit robustness in the encoding phase. A workaround to the above problem can make DAEs an excellent choice for unsupervised learning. DAEs can be a very appealing alternative due to their simplicity for training and elegant interpretation. Due to their simplicity in the training criterion, DAEs have the potential to adapt to different kinds of architectures and to scale with the application. We argue that the current limitation of current DAEs can be overcome by making a very simple modification to the training objective leading to learning better representations in terms of robustness. The modified objective for DAE and is justification are the main contributions of our work.

## 2 DENOISING AUTO-ENCODERS

Let $X$ be the random variable representing the input, and $\hat{X}$ be a corrupted version obtained by an an stochastic operator $q_{\mathcal{C}}(\hat{X}|X)$. The goal of the DAE is to learn a deterministic encoder-decoder composition $f(\cdot) = \tilde{G}^{-1}(G(\cdot))$ such that, for $Y = f(\hat{X})$ the quantity $\mathrm{E}[D(Y, X)]$ is minimized. The corruption process leads to learning non-trivial transformations of the input in particular for the over-complete cases where identity mapping can be learned if there is no appropriate capacity control mechanism in place. Let $G$ and $\tilde{G}^{-1}$ be parametrized by $\boldsymbol{\theta} = \{\boldsymbol{\theta}_{\mathrm{enc}}, \boldsymbol{\theta}_{\mathrm{dec}}\}$. Parameter learning of the DAE can be formulated as the following optimization problem:

$$\underset{\boldsymbol{\theta} \in \boldsymbol{\Theta}}{\operatorname{minimize}} \frac{1}{N} \sum_{i=1}^{N} \mathrm{E}_{\hat{X}_i}[D(f_{\boldsymbol{\theta}}(\hat{X}_i), \mathbf{x}_i)], \tag{1}$$

where $\hat{X}_i \sim q_{\mathcal{C}}(\hat{X}|X = \mathbf{x}_i)$. As we already mentioned, a solution to (1) does not impose explicit restrictions to the encoder. In this approach, only the dimensionality and the range of the encoder can be controlled by modifying its architecture. Therefore, there exist the possibility that the encoder could be learning a feature map, for which some dimensions still carry the effects of perturbing the original input. In many cases, the actual denoising could be taking place in the decoding phase. Below, we present a simple way to overcome the above limitation based on a modification to the DAE objective (1).

### 2.1 MODIFIED DENOISING OBJECTIVE

In order to explicitly enforce robustness in the encoding phase, we can measure the amount of distortion in the encoder by comparing the resulting feature values for the original uncorrupted input against the feature values obtained from encoding its corrupted counterpart. Likewise $D$ in (1), the difference between can be measured by a function $D_{\mathrm{enc}}(\cdot, \cdot)$, for instance cross-entropy can be used for encoder that map input values to $[0, 1]$. This leads to the following modified optimization problem:

$$\underset{\boldsymbol{\theta} \in \boldsymbol{\Theta}}{\operatorname{minimize}} \frac{1}{N} \sum_{i=1}^{N} \mathrm{E}_{\hat{X}_i}[D(f_{\boldsymbol{\theta}}(\hat{X}_i), \mathbf{x}_i)] + \frac{1}{\lambda} \mathrm{E}_{\hat{X}_i}[D_{\mathrm{enc}}(G(\hat{X}_i), G(\mathbf{x}_i))], \tag{2}$$

where $\lambda$ is a tradeoff parameter. The original DAE objective can be simple obtained by setting $\lambda$ to infinity.

### 2.2 ANALYSIS OF THE MODIFIED DAE

The modified objective (2) appears as an intuitive way to explicitly enforce robustness in the encoding phase. Unlike the plain objective where $Y = f(X)$, the stochastic operator $q_{\mathcal{C}}$ in conjunction

with the encoder function $G(.)$ lead to a random map from an instance $\mathbf{x}$ of $X$ to the random variable $Z|X = \mathbf{x}$. In this case, the minimization of the reconstruction error leads to maximization of a lower bound on the mutual information between $Z$ and $X$[1]. However, the information maximization perspective does not provide insights about how the noise acts as capacity control on the encoder. Since the only constraint on the entropy $H(Z)$ is the encoder architecture, mutual information $I(X; Z) = H(X) - H(X|Z) = H(Z) - H(Z|X)$ does not tell us much about the conditional entropy $H(Z|X)$, which is directly related to the encoder. Bear in mind that the corruption process is sill under consideration, thus the map from $X$ to $Z$ is stochastic. The conditional entropy $H(Z|X)$ can be upper bounded by $H(Z|G(X))$[2]:

$$H(Z|X) \leq H(Z|G(X)) \quad = \quad \mathrm{E}_{\hat{X}, G(X)}[-\log(p(G(\hat{X})|G(X)))] \tag{3}$$

$$\propto \quad \mathrm{E}_{\hat{X}, X}[D_{\text{enc}}(G(\hat{X})|G(X))], \tag{4}$$

which corresponds to the population version of the second term of the objective (2).

## 2.3 ARCHITECTURAL CONSIDERATIONS AND THE ENCODING LOSS

The above formulation assumes that the encoding loss term matches the conditional log likelihood fun-ction. Thus, the encoding loss should also match the encoding architecture. For instance, if we use sigmoidal units, we can assume a multivariate Bernoulli distribution of the code and use cross-entropy as the encoding loss. Architectures inducing a continuous code space may require further considerations for the choice of the loss or architectural constraints. For instance, a linear encoder with the euclidean distance as the encoding loss may lead to a trivial minimization of the modified objective. By reducing the norm of the encoder weights, the distance between the codes of the clean and corrupted samples can be made arbitrarily small, only being compensated by scaling of the decoder weights. In this case, a simple way to circumvent this issue is by using tied weights in the decode. In linear case, the role of tied weights is easy to understand. However, a similar interpretation may not extend to non-linear cases. To avoid trivial shrinkage of the encoding space, and without resorting to tied weights, we can employ a normalized distance function such as squared euclidean distance divided by the total variance of the encoded data or by adding a batch normalization layer to the top layer of encoder.

## 2.4 RELATION TO OTHER APPROACHES

In a similar spirit to the proposed modified denoising auto-encoder objective, contractive auto-encoders achieve robustness in the encoder by explicitly computing a regularization term based on the $L^2$ of the Jacobian of the encoder. In the regime of small Gaussian perturbations the modified objective can be approximated by the Jacobian of the encoding transformation. Nevertheless, the setting of the modified denoising auto-encoder is more general in terms of distances and forms of input corruption, which can lead to different properties to the ones obtained by manipulating the Jacobian of the transformation. Another interpretation of the learning algorithm is as a Siamese network of the encoders, where the goal is to map the clean and corrupted input to the "same" code.

## 3 EXPERIMENTS

This section describe some of the experiments we have carried out with the modified DAE. We qualitatively illustrate how in over-complete representations, the DAE benefits from adding the penalty to the encoding space.

## 3.1 SYNTHETIC DATA

**Gaussian distributed data:** The first example corresponds to a set of data point drawn from a bivariate Gaussian distribution with zero mean and covariance matrix

$$\boldsymbol{\Sigma}_X = \begin{pmatrix} 1 & 0.95 \\ 0.95 & 1 \end{pmatrix}. \tag{5}$$

---

[1]Note that the information maximization view applies to conventional auto-encoder, as well. The main difference being the stochastic map

[2]This follows from the data processing inequality.

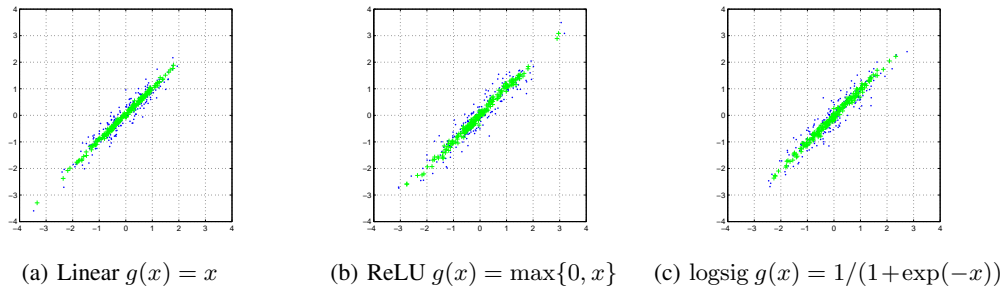

(a) Linear $g(x) = x$  (b) ReLU $g(x) = \max\{0, x\}$  (c) logsig $g(x) = 1/(1+\exp(-x))$

Figure 1: Outputs for different activation functions of the modifed DAE with an over-complete representation when the inputs are Gaussian distributed.

We use a linear encoding, that is $g(x) = x$, that is also over-complete since the encoder project the 2-dimensional data points to 10 different directions. The conventional auto encoder would over fit being able to achieve zero reconstruction error, but it won't be able to implicitly retain what is thought to be the structure in the data. We also compare this output to the outputs of two nonlinear auto-encoders, one uses the logsig units $g(x) = 1/(1 + \exp(-x))$, and the other a rectified linear units (ReLU) $g(x) = \max\{0, x\}$. Figure 1 shows the outputs of the three auto-encoders on the

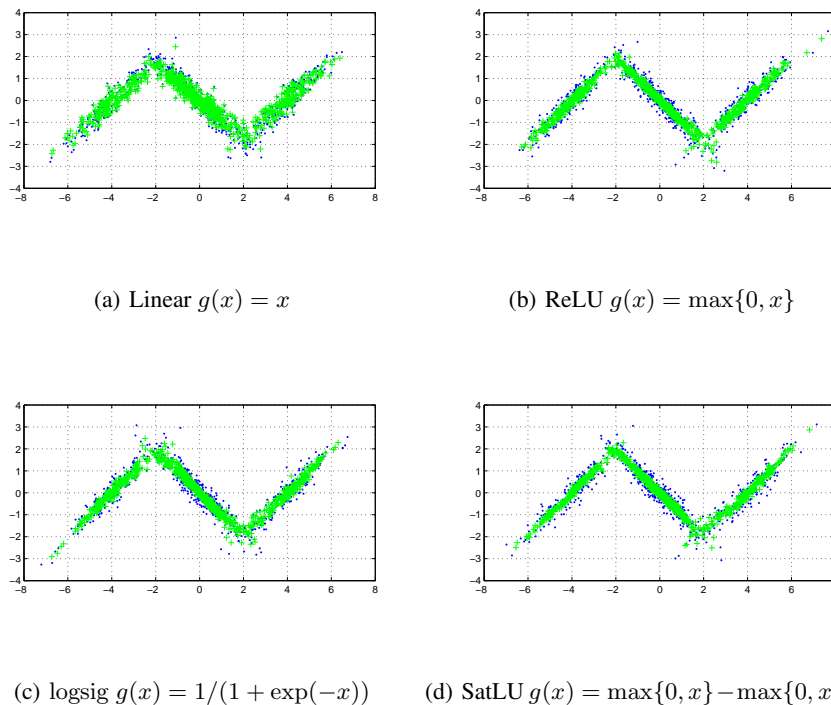

(a) Linear $g(x) = x$  (b) ReLU $g(x) = \max\{0, x\}$

(c) logsig $g(x) = 1/(1 + \exp(-x))$  (d) SatLU $g(x) = \max\{0, x\} - \max\{0, x-1\}$

Figure 2: Outputs for different activation functions of the modified DAE with an over-complete representation when the inputs are a mixture of Gaussian distributions.

Gaussian distributed data. It can be seen that the outputs approximately align with what corresponds roughly to the first principal component of the data. Notice, that no bottleneck neither shrinkage of parameters was explicitly defined. The parameters of our cost function are $\lambda = 1$ for the distortion trade-off, and $\sigma = 0.5$ for the noise level.

**Mixture of Gaussians:** The second example, employs a mixture of three Gaussian distributions to show the output of modified DAE in a nonlinear scenario, where an over-complete representation

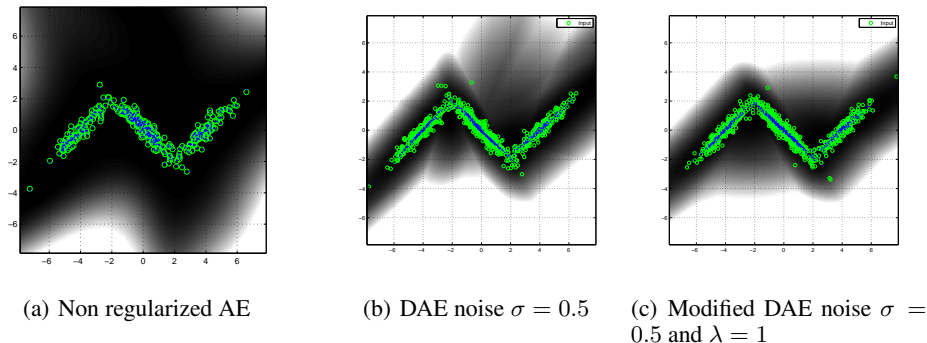

(a) Non regularized AE (b) DAE noise $\sigma = 0.5$ (c) Modified DAE noise $\sigma = 0.5$ and $\lambda = 1$

Figure 3: Energy $\|x - \hat{x}\|^2$ landscapes for different AE algorithms

followed by a nonlinearity can be advantageous. The means of and covariances of the mixture components are

$$
\mu_1 = \begin{pmatrix} 2 \\ -2 \end{pmatrix}, \ \mu_2 = \begin{pmatrix} -2 \\ -2 \end{pmatrix}, \ \mu_1 = \begin{pmatrix} 6 \\ -2 \end{pmatrix}; \text{ and}
$$
$$
\Sigma_1 = \begin{pmatrix} 1 & -0.95 \\ -0.95 & 1 \end{pmatrix}, \ \Sigma_2 = \Sigma_3 = \begin{pmatrix} 1 & 0.95 \\ 0.95 & 1 \end{pmatrix}, \tag{6}
$$

respectively, and the mixing weights are $p_1 = 0.5$, and $p_2 = p_3 = 0.25$.

Figure 2 shows the outputs of the four auto-encoders on the mixture of Gaussian distributions. The auto-encoders employ: linear, rectified linear, sigmoidal, and saturated linear units. It can be seen that the outputs approximately align with what can be though as the principal curves of the data. Again, we want to stress that no bottleneck neither shrinkage was explicitly defined. In this case each of the auto-encoder has 20 units for encoding, which would easily over fit the data in the absence of any regularization or add-hoc constraints such as tied weights. The parameters of our cost function are $\lambda = 1$ for the distortion level $\sigma = 0.5$. The linear units seem to fit the data, but as we previously mentioned they favor the principal components. Increasing the noise would collapse the reconstructed points into a line. This is not necessarily the case when nonlinear units are considered.

Finally, in Figure 3 we show the resulting energy $\|x - \hat{x}\|^2$ landscapes for the over-complete auto-encoder with rectified linear units after being trained with: original DAE objective and the proposed modified DAE objective. The modified objective makes the AE carve well-defined ravines in the energy landscape at the points were majority of the data lies.

## 3.2 MNIST

Here, we observe the influence of the extra term in the objective function of the DAE. We train a single hidden layer DAE with logsig activation and cross entropy loss on both encoding and decoding layers. The number of fully connected hidden units is 2048. Input images are corrupted with zero-mask noise with 30% corruption level. Once the DAE is trained for 100 epochs, we use the encoding layer as a feature extractor for a multi class logistic regression classifier. Figure 4(a), displays the average test error over 30 runs of the DAE training using random initial conditions for different values of $\lambda$. We remind the reader that there is no fine tuning of the weights of the encoder layer, after unsupervised pre-training only the weights of the logistic regression layer are trained using label information. Moreover, since the main goal of here is to observe the influence of the encoding cost term, we focus on comparisons at the single hidden layer level rather than trying to increase performance by stacking multiple layers.

## 4 CONCLUSIONS

We presented an algorithm for learning auto-encoders based on an modified objective for Denoising that explicitly enforces the denoising to be carried out during the encoding phase. Moreover,

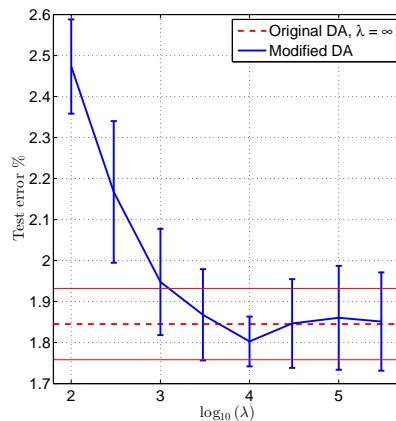

(a) Test errors for different values of $\lambda$

we described how the modified objective can be understood as minimizing an upper bound on the conditional entropy of the code given the inputs assuming the corruption process to be part of the encoding. By simply minimizing the distance between the encoding of the uncorrupted inputs and the encoding of their corrupted counterparts, we show that robustness of the auto encoder can be guaranteed at the encoding phase. Experiments using over-complete bases showed that the modified DAE was able to learn useful encoding mapping (representation) and can also learn a regularized input-output in an implicit manner.

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
