# Peer review of "Revisiting Denoising Auto-Encoders"

_ICLR 2017 — rejected_

[Author Response · Luis Gonzalo Sanchez Giraldo · 15 Dec 2016]
**paper update**

An updated version of the paper taking into consideration the reviewers comments has been uploaded

[Official Review · AnonReviewer2 · rating 5 · confidence 4 · 16 Dec 2016]
**Well motivated, if incremental improvement**

The paper proposes to add an additional term to the denoising-autoencoder objective. The new term is well motivated, it introduces an asymmetry between the encoder and decoder, forcing the encoder to represent a compressed, denoised version of the input. The authors propose to avoid the trivial solution introduced by the new term by using tied weights or normalized Euclidean distance error (the trivial solution occurs by scaling the magnitude of the code down in the encoder, and back up in the decoder). The proposed auto-encoder scheme is very similar to a host of other auto-encoders that have been out in the literature for some time. The authors evaluate the proposed scheme on toy-data distributions in 2D as well as MNIST. Although the work is well motivated, it certainly seems like an empirically unproven and incremental improvement to an old idea.

[Official Review · AnonReviewer1 · rating 4 · confidence 4 · 17 Dec 2016]
**Incremental work without thorough empirical results**

The work introduced a new form of regularization for denoising autoencoders, which explicitly enforces robustness in the encoding phrase w.r.t. input perturbation. The author motivates the regularization term as minimizing the conditional entropy of the encoding given the input. The modifier denoising autoencoders is evaluated on some synthetic datasets as well as MNIST, along with regular auto-encoders and denoising autoencoders. The work is fairly similar to several existing extensions to auto-encoders, e.g., contractive auto encoders, which the author did not include in the comparison.   The experiment section needs more polishing. More details should be provided to help understand the figures in the section.

[Official Review · AnonReviewer3 · rating 4 · confidence 5 · 20 Dec 2016]
**Incremental with too little empirical evidence and insufficiently developed info-theoretic argument.**

The paper proposes a modified DAE objective where it is the mapped representation of the corrupted input that is pushed closer to the representation of the uncorrupted input. This thus borrows from both denoising (DAE) for the stochasticity and from the contractive (CAE) auto-encoders objectives (which the paper doesn’t compare to) for the representational closeness, and as such appears rather incremental. In common with the CAE, a collapse of the representation can only be avoided by additional external constraints, such as tied weights, batch normalization or other normalization heuristics. While I appreciates that the authors added a paragraph discussing this point and the usual remediations after I had raised it in an earlier question, I think it would deserve a proper formal treatment. Note that such external constraints do not seem to arise from the information-theoretic formalism as articulated by the authors. This casts doubt regarding the validity or completeness of the proposed formal motivation as currently exposed.  What the extra regularization does from an information-theoretic perspective remains unclearly articulated (e.g. interpretation of lambda strength?).

On the experimental front, empirical support for the approach is very weak: few experiments on synthetic and small scale data. The modified DAE's test errors on MNIST are larger than those of Original DAE all the time expect for one precise setting of lambda, and then the original DAE performance is still within the displayed error-bar of the modified DAE. So, it is unclear whether the improvement is actually statistically significant.

[Final Decision · Program Chairs · 06 Feb 2017]
**ICLR committee final decision**

The proposed modification of the denoising autoencoder objective is interesting. Shifting some of the burden to the encoder has potential, but the authors need to to show that this burden cannot be transferred to the decoder; they propose some ways of doing this in their response, but these should be explored. And more convincing, larger-scale results are needed.